

# Commonly used methods fail to detect known phase speeds of simulated signals of Sea Surface Height Anomalies

Yair De-Leon[1], Nathan Paldor [1]

[1]Fredy and Nadine Herrmann Institute of Earth Sciences, The Hebrew University of Jerusalem, Edmond J. Safra Campus,
Givat Ram, Jerusalem, 9190401, Israel

*Correspondence to*: Nathan Paldor (nathan.paldor@huji.ac.il)

**Abstract.** This work examines the accuracy and validity of two variants of Radon transform and two variants of the Two-Dimensional Fast Fourier Transform (2D FFT) that have been previously used for estimating the propagation speed of Sea Surface Height Anomalies (SSHA) derived from satellite borne altimeters. The examination employs numerically simulated SSHA signals made up of 20 or 50 modes where one, randomly selected, mode has a larger amplitude than the uniform amplitude of the other modes. Since the dominant input mode is ab-initio known, we can clearly define "success" in detecting its phase/propagation speed. We show that all previously employed variants fail to detect the phase speed of the dominant input mode even when its amplitude is 5 times larger than all other modes and that they successfully detect the phase speed of the dominant input mode only when its amplitude is 10 times (or more) larger than the other modes. This requirement is an unrealistic limitation on oceanic SSHA observations. In addition, three of the variant methods "detect" a dominant mode even when all modes have the exact same amplitude. The accuracy with which the four algorithms identify a dominant input mode **decreases** with the increase in the number of modes in the signal. Our findings cast a doubt on the reliability of phase speed estimates of SSHA observations and the reported "too fast" phase speed of baroclinic Rossby waves in the ocean.

## 1 Introduction

Satellite observations of Sea Surface Height Anomalies (SSHA) were routinely conducted since the early 1990's in many parts of the world ocean by various satellite borne altimeters. With the advance of technology, current data are obtained using up to 4 satellites simultaneously which yields spatial resolution of 1/4° of latitude and longitude and temporal resolution of 1 day. Analyses of observations of SSHA at any point in the ocean have shown that these features propagate westward only which led to their interpretation as either surface manifestations of Rossby waves that propagate westward or the propagation of mesoscale eddies (whose propagation speeds are similar to the phase speeds of Rossby waves).

The rate at which SSHA propagate westward can be derived using time-longitude (Hovmöller) diagrams at certain latitude where the inverse slopes of same-amplitude contours are proportional to the propagation speed of these contours, i.e. proportional to the propagation speed of Rossby waves/eddies. These slopes were calculated by applying various methods, employed in image processing and detailed below in Sect. 2.2, to the data distributed by AVISO (supported by CNES; see



http://www.aviso.altimetry.fr/duacs/) or to processed data (e.g. Polito and Liu, 2003 who separated the data into tiles of different periods).

The phase speed estimates from observations were compared with theoretical predictions of the classical harmonic theory of Rossby wave (e.g. Pedlosky, 1987; Gill, 1982) and it was found that the observed phase speed exceed the theoretical one in many locations in the world ocean (e.g. Chelton and Schlax, 1996; Osychny and Cornillon, 2004). Explanations for these discrepancies were suggested in previous studies (see the introduction of De-Leon and Paldor, 2017b for more details) in order to bridge that gap. One of the suggestions is the trapped wave theory (Paldor, Rubin and Mariano, 2007; Paldor and Sigalov, 2008; Gildor et al., 2016) where the waves' phase speed is higher than that of harmonic waves. However, this trapped wave theory could be applied and compared to oceanic observations only in the presence of a wall, while there are only a few such places in the world (the southern coast of Australia is the only place where we succeeded to do that, see De-Leon and Paldor, 2017b). An attempt to define a virtual boundary based on a linear fit of observed estimates have not yielded satisfactorily fit between the trapped wave theory and observations.

A possible reason for this lack of success in bridging the gap between observations and theory is that the observed estimates are not reliable. De-Leon and Paldor (2017a) examined the accuracy of various methods in estimating the phase speed of waves by applying these methods to an artificially generated signal made of 3 sine functions (modes) with known phase speeds and amplitudes, compounded by large-amplitude random white noise. All methods have successfully filtered out the high amplitude white-noise from the 3-harmonic signal and accurately detected the main mode (some of them also detected the secondary modes). However, such a signal is too synthetic/ideal and cannot be compared to real oceanic observations that include tens, if not hundreds, of modes with different frequencies and phase speeds and not just 3 modes compounded by white noise.

In this short study we simulate "oceanic observations" and examine whether the methods detect a single dominant phase speed out of many (20 or 50) phase speeds. In Sect. 2 we provide details on the generation of "observed" signal, the methods for evaluating the phase speed of the signal and the tests we apply to the signals. The results are shown in Sect. 3 and discussed in Sect. 4.

## 2 "Data" and methods

### 2.1 Generating the simulated "observations"

The sea surface height "observations" used here are generated numerically by summing up N purely propagating sine functions (modes, hereafter) of the form $\sin(kx-\omega t)$ where $k$ is the zonal wavenumber, $x$ is longitude, $t$ is time and $\omega=kC$ is the frequency (where $C$ is the zonal phase speed). The number of participating modes, N, is taken to be either 20 or 50 and N-1 of these modes have an amplitude of 1 while the amplitude of the additional $N^{th}$, randomly chosen, mode is either larger than or equal to 1. The sum of all N modes constitutes the signal which is analyzed by the methods described in 2.2.





The "spatial domain" (*x*) is chosen between "longitudes" 70-130° on a Cartesian grid with a 1/4° resolution, and the "time" (*t*) duration is 20 years (1044 weeks) with temporal resolution of one datum per week. The values of the phase speeds, *C*, are uniformly distributed in the 0 to -18 $\frac{cm}{s}$ range, which is typical for baroclinic Rossby waves in the ocean in mid-latitudes (see e.g. Chelton and Schlax, 1996). The values of the frequencies, $\omega$, are selected randomly so that the period,

$2\pi/\omega$, falls in the range between 5 and 200 weeks while the values of the zonal wavenumbers, *k*, equal $\omega/C$. The resulting signal was low-pass filtered by applying a 5-week-running-average at each grid point to eliminate short term variations.

The SSHA signal made up of the filtered signal (i.e. the sum of N pure sine waves) at a given latitude, is plotted as a function of longitude and time (AKA the Hovmöller diagram). When a single dominant mode exists that has a certain phase speed the pattern on this diagram is a straight line whose slope is the inverse of the dominant phase speed (since the abscissa

is longitude and the ordinate is time). An example of a time-longitude diagram of an artificial signal is shown in Fig. 1a (for signal with dominant input mode's amplitude of 1.5) where the slope of the solid blue line corresponds to the phase speed of the dominant input mode. The challenge is to estimate the dominant speeds using different methods and examine their success in detecting the phase speed of the known dominant input mode. The methods examined here are detailed in the next subsection.

**2.2 Methods for estimating the "observed" phase speed**

Four variants of methods have been employed for identifying the preferred direction of the same-amplitude contours on the Hovmöller diagram, each variant relates a certain measure of the power/intensity in a mode to its phase speed. The first method is the Radon transform, used by e.g. Chelton and Schlax (1996); Chelton et al. (2003) and Tulloch et al. (2009) for analyzing satellite observations of the ocean. In this method, one calculates the sum of the amplitudes along lines inclined at

an angle $\theta$ and displaced a distance *s* from the origin. Then, the sum of squares of the values of these sums along all lines having the same angle is calculated and the angle at which this sum of squares is maximal, is the best estimate for the orientation of the lines on the image. The dominant phase speed of the signal is then proportional to the tangent of this angle of maximum sum-of-squares. In the second method, which is a variant of the Radon transform, the variance of the amplitudes is calculated along every angle $\theta$ instead of the sum of amplitudes. This method was used e.g. by Polito and Liu

(2003) and Barron et al. (2009). Another, independent, method commonly used (e.g. Zang and Wunsch, 1999; Osychny and Cornillon, 2004) is the two-dimensional Fast Fourier Transform (2D FFT). An example of ($\omega$, *k*) diagram, obtained by applying 2D FFT to the signal of Fig. 1a is shown in Fig. 1b. Here we use two variants of the 2D FFT method: In the first variant one sweeps over the 2D FFT spectra to find the direction in ($\omega$, *k*) plane with maximum "energy" (this is the third method) while in the second variant one finds the maximal amplitude of the 2D FFT (i.e. one of the bright points in Fig. 1b)

and calculates the ratio $\omega/k$ where $\omega$ and *k* are the frequency and zonal wavenumber of the maximal amplitude (this is the fourth method). Detailed description of these methods and the interpretation of observed signals are found in De-Leon and Paldor (2017a).



Each of the four variants of the methods can yield an estimate of the dominant phase speed of a signal, based on the extremum of a graph that relates the calculated measure of a mode's intensity to its phase speed. An estimation of the phase speed based on a local extremum of this graph is accepted when this extremum is narrow and isolated compared to other local extrema. When the normalized amplitude (the term normalized amplitudes is used for the calculated amplitudes divided

by the maximal amplitude in the domain) of a distinct peak is 1 while the (normalized) amplitudes of all other peaks are smaller than 0.8 this mode is considered the dominant mode. In the variance method, where the extrema are minima, the dominant mode is accepted when its amplitude is 0.0 while the normalized amplitudes of all other minima are larger than 0.2. The results shown in Fig. 2 demonstrate the emergence of a "rejected" peak that does not differ significantly from other peaks (2d) and "accepted" peaks whose intensities are significantly larger than those of other peaks (2a, 2c) or "accepted"

trough that is significantly smaller than the other troughs (2b).

## 2.3 Examining the accuracy of dominant mode detection

Two types of tests are applied to these "observations" to examine the accuracy of the various methods in assessing the existence of a dominant phase speed (i.e. mode) in the signal.

The first is a true-positive/false-negative test in which there is a dominant mode in the "observed" signal and a given

method indicates whether this dominant mode exists (true-positive; TP) or not (false-negative; FN). In our case, one of the sine functions (this is the dominant input mode) is chosen randomly and its amplitude is set to be larger than 1. We check for each method if (at all) it identifies a dominant mode and if so, if it matches the phase speed of this (larger amplitude) input mode. We divide the interval of phase speeds into N (20 or 50) bins of equi-distant values and the determination of the success of the methods in identifying a dominant mode is as follows: if the dominant mode found by the method falls in the

expected bin of the dominant input mode - we score it by 1 ("TP"). If it is found in one of its next neighbors, it is scored by 1/2. A score of 0 ("FN") is assigned when the method cannot find any dominant mode and when it detects a dominant mode more than 1 bin away from the correct bin. For each of 5 values of dominant mode's amplitudes: 1.5, 2, 2.5, 5 and 10 and for 2 values of N (20 or 50) we repeat this procedure 50 times (i.e. for 50 different signals), sum up the scores and calculate the percentage of success in identifying the dominant input mode in the signals by TP/(TP+FN)*100.

The second is a false-positive/true-negative test in which no dominant mode exists in the "observed" signal and a given method indicates that there **exists** a dominant mode (false-positive; FP) or not (true-negative; TN). In our case, this is done by generating a signal in which all modes have identical amplitudes (=1) and checking whether a method erroneously detects a certain phase speed as dominant. If a dominant mode is detected, we score it by 1 ("FP"), if a peak is detected but is too wide we score it by 1/2 and if there is no dominant mode (i.e. no distinct peak or more than one peak) we score it by 0

("TN"). We repeat this procedure 50 times for each of N=20 or N=50, sum up the scores and calculate the percentage of erroneous detection of dominant mode in the signals by FP/(FP+TN)*100.



## 3 Results

An example of erroneous determination of the dominant mode is shown in Fig. 2 for the signal shown in Fig. 1a. Figure 2a shows the distribution of the sum-of-squares of the Radon transform versus $C$ (black markers), normalized such that the maximum value equals 1. Also plotted are the dashed black vertical lines located at the N values of the uniformly distributed

phase speeds, $C$, where the solid blue line is located at the $C$-value of the dominant input mode's phase speed. These dashed black and solid blue vertical lines are also shown in panels (b)-(d) of Fig. 2. Clearly, the dominant phase speed calculated by the Radon transform (where the black curve attains its maximum) does not match the phase speed of the dominant input mode. Figure 2b shows the (normalized) mean of variances as a function of $C$ (black markers), and here, too, the calculated phase speed (the curve's minimum point) does not agree with the dominant input mode's speed. Figure 2c shows the

(normalized) distribution of the sum-of-squares of the spectral coefficients (2D FFT-amplitudes) along different $\omega/k$ lines (sweeping) as a function of $C$ (black markers) where a distinct peak exists but it's located far from the dominant input mode's phase speed. Figure 2d shows the 20 highest (normalized) 2D FFT amplitudes of the $(\omega, k)$ diagram of Fig. 1b. There are many peaks with no clear single maximum and the dominant input mode has one of the lowest amplitudes. For this signal, none of the methods identified correctly the dominant input mode.

The statistics of success of each method in detecting the dominant input mode for the TP/FN test is shown in Fig. 3 for dominant input mode's amplitude of 2.5, 5 and 10. The results for amplitudes of 1.5 and 2 are not shown as the success rate of all methods at these amplitudes is between 10% and 30% only. In each case we repeated the procedure 50 times (i.e. for 50 signals) for N=20 and then for N=50, summed the scores and calculated the percentage of success by TP/(TP+FN)*100. The conclusions from these results are: 1) in order to identify the dominant input mode with more than 70% certainty, its

amplitude should be larger than 5. 2) No method has clear advantage over the other methods. 3) Clearly, as N increases the dominant mode's amplitude has to increase, too, for a successful identification (so as to ensure that the ratio between the dominant mode's amplitude and the sum of all amplitudes is similar for different values of N since, e.g., 2.5/20>2.5/50).

    The statistics of erroneous detection of a dominant mode (the FP/TN test where the amplitudes of all input modes equal 1 i.e. there is no dominant input mode) is shown in Fig. 4 for each of the methods (here again we have generated 50 signals

for N=20 and for N=50 and calculated the percentage of erroneous detection by FP/(FP+TN)*100. The 2D FFT maxima method is the only method for which the percentage of error is smaller than 20% while other 3 methods err in at least 50% of the cases (i.e. they identify a single clear peak in one of the phase speeds). As N increases this erroneous detection percentage decreases slightly in the 2 Radon variants but increases slightly in the 2D FFT sweeping method.

## 4 Discussion

All methods cannot identify a dominant input mode unless its amplitude is significantly larger than the others (by a factor larger than 5!) and most of them (except the 2D FFT maxima) erroneously detect a dominant mode when there is no such input mode. Though the 2D FFT maxima method does not falsely detect dominant mode when it does not exist, its



performance in detecting dominant input mode when it exists is not satisfactory. For realistic signals of the ocean we don't know that there is a dominant mode with sufficiently large amplitude so none of the methods is reliable for estimating the phase speed of Rossby waves.

When the values of $\omega$ and $k$ are chosen in a range corresponding to the resolution limit and the Nyquist frequency, the
success of 2D FFT in identifying the dominant input mode's phase speed increases significantly, compared to the case where the values of $C$ are determined in advance, $\omega$ is chosen randomly in a particular range and $k$ is set as a result of $\omega/C$ so aliasing can occur. For that reason, even if the signal includes only one mode (i.e. one $C$), and both $\omega$ and $k$ are chosen in the latter manner, there can be a wrong identification, and therefore this method does not succeed by more than 70% even if the amplitude of the dominant input mode is 10 times that of the other amplitudes. In the ocean we don't know ab-initio which
wave numbers and frequencies exist so we cannot filter them out of the signal; hence aliasing can occur, and the percentage of success in detecting the real dominant mode is expected to decrease further.

The erroneous identification of the Radon and variance methods can be partially attributed to the non-linear relation between the angle $\theta$ and the phase speed, $C$, which is proportional to $\tan(\theta)$ so the equi-distant values of $C$ are converted to $\theta$-values that are very close to one another. Figure 5 shows the distribution of the sum of squares of the Radon transform
versus $\theta$ for the signal shown in Fig. 1a (where the distribution of the sum of squares of the Radon transform versus $C$ for that signal is shown in Fig. 2a). It is clear from this figure that the peak is located in the vicinity of $\theta$ values corresponding to many $C$ values. However, the performance of the Radon variants improves with the increase of the dominant input mode's amplitude, so the non-linear relation between the angle and phase speed is not the only reason for the mismatch.

As N (the number of modes) increases (it is impossible to establish a-priori a bound on the number of modes in the
ocean), the dominant input mode's amplitude should be larger in order to be separately identified from modes with similar characteristics. In addition, the width of the bins becomes narrower as N increases so fewer results can be evaluated as success.

The weakness of the methods/algorithms in identifying the dominant mode points to the difficulty in comparison between theories and observations and this difficulty might explain the lack of "continuity" of phase speed estimates between
adjacent latitudes in one (or more) methods. It can also explain why a validation of the higher order trapped wave theory (where the $\beta$ term is treated consistently) has been confirmed by observations only in the Indian Ocean south of Australia (De-Leon and Paldor, 2017b) and not in other parts of the world ocean.

*Competing interests*. The authors declare that they have no conflict of interest.



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




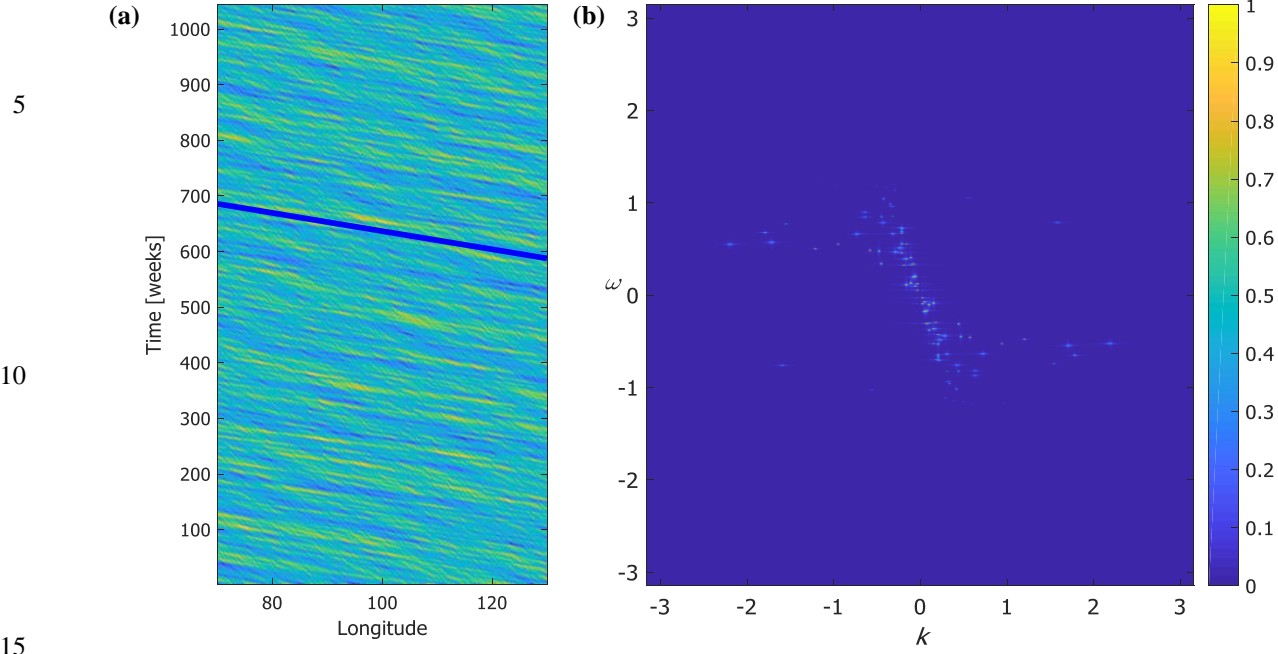

**Figure 1: (a)** An example of an artificial "observed" signal. **(b)** The associated $(\omega, k)$ diagram obtained by applying 2D FFT to the signal of panel (a). The solid blue line in panel (a) corresponds to the randomly chosen dominant input mode's phase speed.





**Figure 2: An application of the 4 methods to the artificially generated signal shown in Fig. 1a (panel a: Radon Transform, panel b: Variance, panel c: 2D FFT sweeping, panel d: 2D FFT maximal amplitude). Blue lines correspond to the dominant input mode phase speed and dashed black lines correspond to the input modes' phase speeds.**

25



**Figure 3: The percentage of success of each variant of the methods for dominant mode's amplitude of 2.5 (panel a), 5 (panel b) and 10 (panel c) for both N=20 and N=50.**



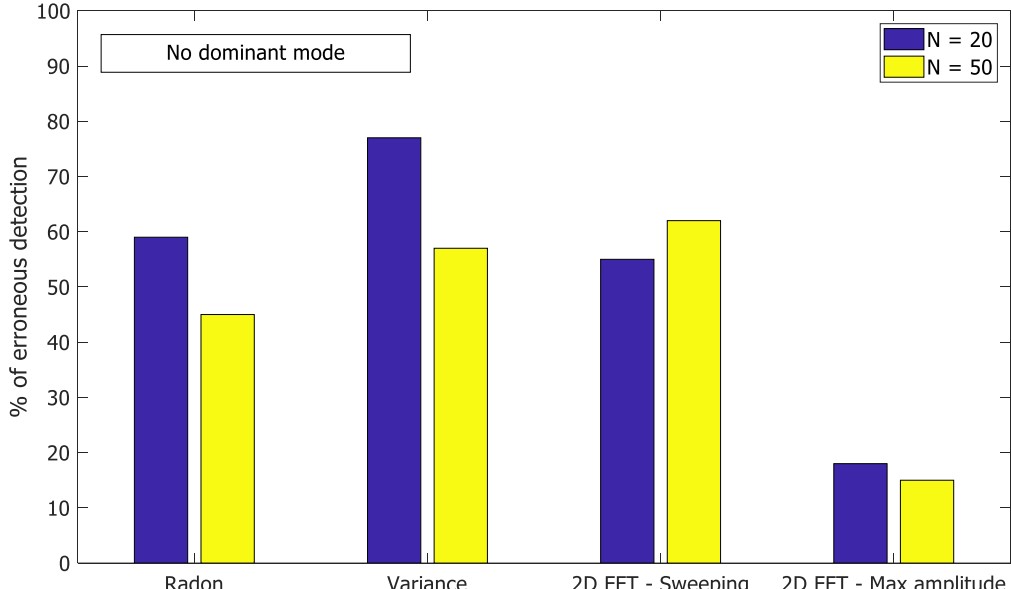

**Figure 4: The percentage of error detection of each variant of the methods where there is no dominant input amplitude.**





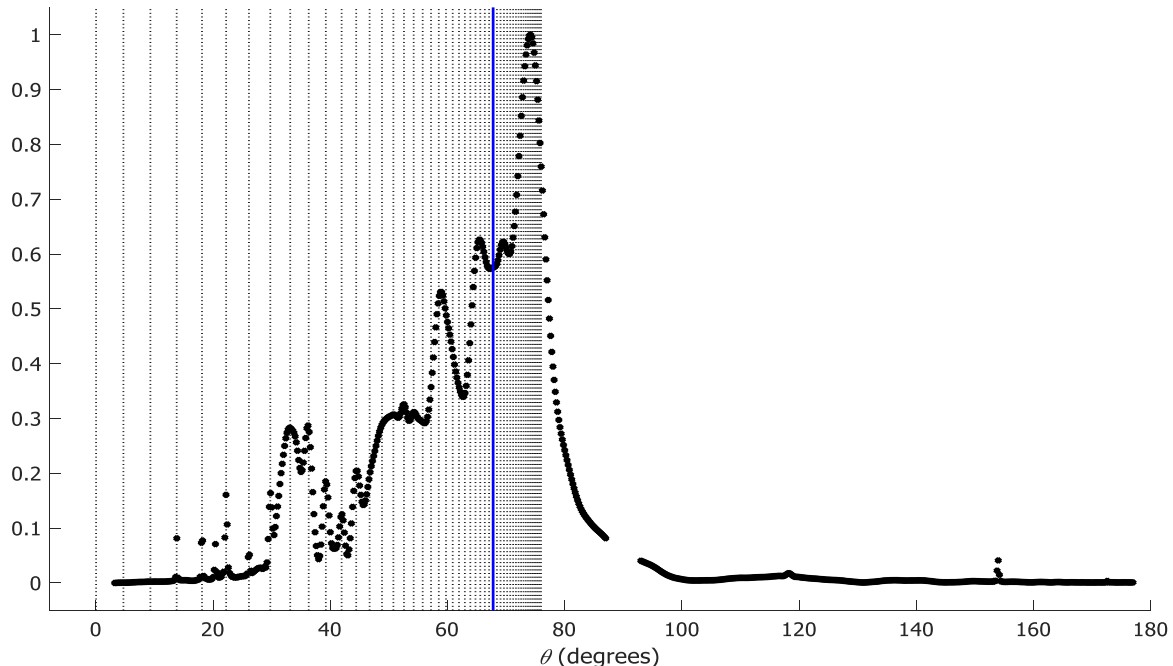

**Figure 5: The distribution of the sum-of-squares of the Radon transform versus the Radon angle $\theta$. Clearly, the distribution of the corresponding input phase speeds (vertical dashed black lines) as a function of $\theta$ is not uniform due to the nonlinear relation**
5 **between $C$ and $\theta$.**