# Peer review of "Commonly used methods fail to detect known propagation speeds of simulated signals from time-longitude (Hovmöller) diagrams"

_Ocean Science, 2019_

## Referee Comment (RC1) · Remi Tailleux (Referee) · 3 Jul 2019

**Summary and recommendation:**

The main aim of this paper is to challenge the reliability of the observational basis for the 'too-fast' Rossby waves evidenced by Chelton and Schlax (1996) based on 4 years of Topex-Poseidon satellite altimeter data. The authors derive their conclusion from showing that it is possible to construct a synthetic Rossby wave signal composed of 20 to 50 sine waves with random known speeds, which standard techniques such as the Radon and Fourier transforms fail to identify accurately. In a previous study, Paldor et al. had showed such techniques to work well for a synthetic signal composed of three

basic waves only, so the difficulties experienced by the Radon and Fourier transforms in this paper appear to result from the increase in many more basic waves in the synthetic signal constructed. As to the motivation for the present study, Nathan Paldor's group has been working on the 'too-fast' Rossby wave issue for many years, promoting the view that the observed phase speed enhancement results from latitudinal trapping due to Earth's curvature. So far, however, Paldor's group appear to have found it difficult to vindicate their theory from observations; but rather than concluding that the problem might rest with their theory, as others theoreticians may have done, the present study proposes that the blame should lie with the observations and the kind of techniques used to analyse them instead, not their theory.

As far as presentation is concerned, the paper is clearly written, and the analysis appears to be competently done. However, as a contribution to the general issue of what satellite altimeter data actually tell us about westward propagation in the ocean and about the usefulness/validity of the standard Rossby wave theory, this study appears to be very biased in its approach and therefore of very little scientific value, clearly failing to meet the required standards for publication. This is unfortunate, because I otherwise find Paldor's work on the rigorous analysis of the waves supported by the shallow water equations to useful and valuable. As far as I understand the issue, their work appears to be essentially concerned with refining the standard flat-bottom, no mean flow, linear theory of the shallow-water waves on the sphere, and has therefore no bearing with real Rossby waves, which theoretical advances over the past 50 years have clearly showed to be strongly affected by both the background mean flow and topography. The rationale for my assessment is contained in the following remarks and observations.

**Main points**

1. The authors fail to mention that the reliability of Chelton and Schlax (1996)'s conclusions has already challenged by Dudley Chelton himself and his collaborators in Chelton et al. (2011), in which the authors argue that westward propagation in the oceans is dominated by meso-scale eddies rather than linear Rossby waves in contrast to what CS96 had previously assumed. Since then, how to disentangle the meso-scale eddy field from the background Rossby wave field has been a major challenge that only a few authors have tried to tackle. Since we know that meso-scale eddies tend to have an equatorward or poleward drift depending on whether they are cyclonic or anti-cyclonic, it is clear that determining their propagation characteristics cannot be easily achieved from the use of Hoevmuller diagrams in longitude/time, which is why eddy tracking algorithms have been developed. Since we don't really know to what extent the propagation speed of eddies differs from that the more linear background Rossby wave field, it seems clear that there is some degree of uncertainty about how CS96's results should be interpreted. In any case, it is clear from Chelton et al. (2011) that there is no observational basis for their synthetic signal.

2. Theoretical developments prompted by Chelton and Schlax (1996) have clearly revealed that the background mean flow and bottom topography have a major impact on the propagation and vertical structures of Rossby waves, and hence that the standard theory can never be a satisfactory description of actual Rossby wave propagation regardless of what satellite altimeter data actually tell us. Indeed, Aoki et al. (2009) and Hunt et al. (2012) have both convincingly established that the standard theory cannot account for the features of simulated Rossby waves propagation, which can only be satisfactorily explained when both the mean flow and bottom topography are accounted for. Flat bottom, no mean flow, modes are completely unable to capture the vertical structure of simulated Rossby wave variability. Irrespective of what the observations tell us, I believe it is pretty clear that the authors' approach cannot tell us anything about actual Rossby waves.

3. Contrary to what this paper and previous ones assert, theoretical studies of the

standard theory based on the WKB approximation are able to account for both the trapping of the Rossby waves as well as for Earth curvature, and it is misleading to refer to such theories as harmonic theories. In WKB theory, one will typically express the pressure anomaly in the form

$$p = A(x, y, z, t)e^{(i\Sigma(x,y,t,))}$$

$$k = \nabla\Sigma, \qquad \omega = -\frac{\partial\Sigma}{\partial t}$$

In such an approach, the amplitude is slowly varying, and will in general decay with latitude, thus capturing the trapped wave behaviour emphasised by the authors. The function $\Sigma$ is a rapidly varying phase function, allowing to define a local wave vector and frequency. Note that a single WKB wave mode is able to represent the observed beta-refraction and a latitudinally varying phase speed. In contrast, the basic wave mode considered by Paldro's group is separable in latitude, and typically chosen of the form

$$p = A(y)e^{i(kx-\omega t)}$$

Arguably, if the term 'harmonic mode' needs to be used, it seems more appropriate to the modes considered by Paldor's group, since it is clearly what they chose for the temporal and zonal dependence of their mode. As a result, such a mode does not capture the beta-refraction pattern described by Shopf et al. (1981) for instance, raising the question of how useful this kind of mode is to describe mid-latitude Rossby waves.

**References:**

Aoki et al., 2009:. Mid-latitude Rossby waves in a high-resolution numerical model simulation. JPO, 39, 2264-2279.

Hunt et al. 2012: The vertical structure of oceanic Rossby waves: a comparison of high-resolution model data to theoretical vertical structures. Ocean Sciences,
8, 19-35.

Please also note the supplement to this comment:
https://www.ocean-sci-discuss.net/os-2019-34/os-2019-34-RC1-supplement.pdf

---

## Referee Comment (RC2) · Anonymous Referee #2 · 8 Jul 2019

Characterizing the properties of oceanic Rossby waves is central to understanding the role of the ocean in the climate as much of the response of the ocean to large-scale forcing is mediated by these waves. Indeed, this issue has attracted considerable attention across the ocean sciences, particularly since the advent of accurate altimetry measurements in 1992 when it became possible, in principle, to observe the signature of Rossby waves at the ocean surface, yet many aspect of such waves remain poorly understood. In particular, it has been found that observed phase speeds derived from altimetry data are systematically faster than the speeds suggested by the theory of Rossby waves. A number of explanations for the disagreement between observations and theory have been proposed, including the effects of the mean zonal flow and bot-

tom topography or the fact that many of the westward-propagating features observed in the altimetry data are, in fact, eddies rather than Rossby waves.

The present study tests the ability of several methods to estimate the phase speed of Rossby waves on simulated data, and finds that such methods very often fail to estimate the true phase speed. The authors then conclude that this is the most likely reason for the differences between observed and theoretical phase speeds. The paper is well written, the figures are mostly adequate and clear, and the experiments designed to assess the skill of the various detection appear to have been conducted appropriately. Unfortunately, although the overall aim of the paper is worth pursuing, some aspects of the paper raise doubts and I do not believe that the results from the performed experiments support the authors' conclusion that "none of the methods is reliable for estimating the phase speed of Rossby waves in the real ocean". The authors are right in concluding that none of the methods is able to estimate the true phase speed in the simulated data, but this conclusion cannot be extrapolated to the observed data since, to the extent that I understand the issue, I don't think the simulated data provides an accurate representation of Rossby waves in the real ocean. In conclusion, I think that the manuscript requires substantial revisions and thus I cannot recommend it for publication as it stands. Details on my main concerns and other minor points are provided below.

Main points:

1. It is unclear to me from Section2.1 how exactly the simulated data are generated. The authors state that "The values of the phase speeds, C, are uniformly distributed in the 0 to -18 cm/s range". Does that mean that each of the 20 or 50 modes is assigned a different phase speed within that range? Long Rossby waves in the ocean are approximately non-dispersive and so their phase speed is the same at all frequencies. Hence, assigning a different speed to every mode, if this is indeed what is done here, seems unjustified. Could you please clarify how exactly phase speed are ascribed to each mode? How do the results change if the same phase speed is used for all modes?

Also, the range -18 to 0 cm /s contains some rather extreme values, do you get the same results if the speeds are generated from the range (-10, -2) cm/s?

2. On a similar comment, the theory of Rossby waves indicates that Rossby waves have a maximum frequency, which for the ocean is quite restrictive. For example, no baroclinic Rossby waves with periods shorter than 13 weeks are possible poleward of about 15o latitude. Here, the periods are taken from the range 5 to 200 weeks, which again seems to include some rather extreme values. Could you please provide a reference supporting such high frequencies for observed Rossby waves? How do the results change if you restrict the periods of the Rossby waves to, for example, the range 15 to 100 weeks?

3. Theoretical phase speeds are not only different from observations, they are systematically slower. If the simulated data were an accurate representation of the real ocean and the detection methods were really the issue here, then the authors should also find a systematic bias in the estimated phase speed. However, there is no mention of this in the paper. The authors only state that all methods fail to estimate the true phase speed of Rossby waves. Do you find any systematic biases? Could you please further elaborate on this?

4. In assessing the skill of the various methods, the authors assign a score of 1/2 if the dominant mode falls in one of its nearest neighbors. This seems to me like a rather arbitrary choice. Why not the second nearest neighbor or the third one? Can you estimate a "standard error" for the phase speed estimates based on the multiple realizations and assign a score of 1 when the true value is within one standard error and zero otherwise? This would be, in my view, a fairer metric for skill. Also, I think that 50 realizations is not sufficient and would suggest you use at least 100, if not 1000.

Minor points:

Page 1. The spatiotemporal resolutions quoted here for the altimetry data refer to the grid size and time step of the altimetry gridded products rather than the scales

that can actually be resolved by altimeters. Depending on latitude, the spatial separation between altimetry tracks can be of several hundred kilometers and altimeters take measurements over the same location once every 10 days at most. I think that some clarification is needed here, along with some references.

Page 1. "these features propagate..." What features? Please clarify.

Page 1. "Rossby waves that propagate westward" I suggest you remove "that propagate westward" as this seems redundant in this particular sentence.

Page 1. replace "diagrams at certain latitude" with "diagrams at a certain latitude".

Page 2. "phase speed exceeds".

Page 2. I suggest "in the -18 to 0 cm/s range".

Page 5. I suggest "None of the methods can identify a dominant input ..."

———————————————

---

## Author Comment (AC1) · 25 Jul 2019

**Author's response to:* Referee #1's comments on "Commonly used methods fail to detect known phase speeds of simulated signals of Sea Surface Height Anomalies" *by* Y. De-Leon and N. Paldor**

**Summary and recommendation:**

The main aim of this paper is to challenge the reliability of the observational basis for the 'too-fast' Rossby waves evidenced by Chelton and Schlax (1996) based on 4 years of Topex-Poseidon satellite altimeter data. The authors derive their conclusion from showing that it is possible to construct a synthetic Rossby wave signal composed of 20 to 50 sine waves with random known speeds, which standard techniques such as the Radon and Fourier transforms fail to identify accurately. In a previous study, Paldor et al. had showed such techniques to work well for a synthetic signal composed of three basic waves only, so the difficulties experienced by the Radon and Fourier transforms in this paper appear to result from the increase in many more basic waves in the synthetic signal constructed. As to the motivation for the present study, Nathan Paldor's group has been working on the 'too-fast' Rossby wave issue for many years, promoting the view that the observed phase speed enhancement results from latitudinal trapping due to Earth's curvature. So far, however, Paldor's group appear to have found it difficult to vindicate their theory from observations; but rather than concluding that the problem might rest with their theory, as others theoreticians may have done, the present study proposes that the blame should lie with the observations and the kind of techniques used to analyse them instead, not their theory.

We appreciate the concise summary the reviewer has written about Paldor's work in the last decade but neither the theoretical work itself nor the reviewer's summary have anything to do with the work under review that examines the applicability of Radon Transform and 2D-FFT methods to time-longitude (Hovmöller) diagrams. We share the reviewer's frustration with the minute impact that a higher-order theory that **consistently** accounts for the latitudinal variation of Coriolis parameter (instead of the traditional paradigm that "f is constant though its derivative is non-zero") had in planar GFD (not only spherical as the reviewer erroneously claims!).

Since no additional assumptions or approximations are employed in the Trapped wave theory (in comparison to the Harmonic traditional theory), and only higher order terms are consistently included, we see no basis for the claim: "… that the problem might rest with their theory". The reviewer is invited to refute the Trapped wave theory in another forum.

As far as presentation is concerned, the paper is clearly written, and the analysis appears to be competently done. However, as a contribution to the general issue of what satellite altimeter data actually tell us about westward propagation in the ocean and about the usefulness/validity of the standard Rossby wave theory, this

study appears to be very biased in its approach and therefore of very little scientific value, clearly failing to meet the required standards for publication. This is unfortunate, because I otherwise find Paldor's work on the rigorous analysis of the waves supported by the shallow water equations to useful and valuable. As far as I understand the issue, their work appears to be essentially concerned with refining the standard flat-bottom, no mean flow, linear theory of the shallow-water waves on the sphere, and has therefore no bearing with real Rossby waves, which theoretical advances over the past 50 years have clearly showed to be strongly affected by both the background mean flow and topography. The rationale for my assessment is contained in the following remarks and observations.

Again, the current work does not deal with the consistent wave theory of Rossby waves (on a sphere or a plane) but with methods for extracting propagation speeds from slopes of contour levels on time-longitude (Hovmöller) diagrams. In our view, the reviewer's assessments: 1) that the paper is "clearly written" and 2) that the analysis is "competently done" along with the prevalent usage of these methods in recent (see the response below to main point #1) interpretations of various oceanic observations should render the paper suitable for publication in Ocean Science.

**Main points**

1. The authors fail to mention that the reliability of Chelton and Schlax (1996)'s conclusions has already challenged by Dudley Chelton himself and his collaborators in Chelton et al. (2011), in which the authors argue that westward propagation in the oceans is dominated by meso-scale eddies rather than linear Rossby waves in contrast to what CS96 had previously assumed. Since then, how to disentangle the meso-scale eddy field from the background Rossby wave field has been a major challenge that only a few authors have tried to tackle. Since we know that meso-scale eddies tend to have an equatorward or poleward drift depending on whether they are cyclonic or anti-cyclonic, it is clear that determining their propagation characteristics cannot be easily achieved from the use of Hoevmuller diagrams in longitude/time, which is why eddy tracking algorithms have been developed. Since we don't really know to what extent the propagation speed of eddies differs from that the more linear background Rossby wave field, it seems clear that there is some degree of uncertainty about how CS96's results should be interpreted. In any case, it is clear from Chelton et al. (2011) that there is no observational basis for their synthetic signal.

We changed the focus of the paper from satellite derived SSHA signals to propagation speeds derived from time-longitude diagrams (but we cannot ignore the simple fact that the Radon transform and 2D FFT methods were heavily employed in SSHA signals derived from satellites). Both the Hovmöller diagrams and the methods employed to interpret them were used in recent years (last 5-6 years) and not only prior to 2011. Additional such references will be included in a revised version of the manuscript.

2. Theoretical developments prompted by Chelton and Schlax (1996) have clearly revealed that the background mean flow and bottom topography have a major impact on the propagation and vertical structures of Rossby waves, and hence that the standard theory can never be a satisfactory description of actual Rossby wave propagation regardless of what satellite altimeter data actually tell us. Indeed, Aoki et al. (2009) and Hunt et al. (2012) have both convincingly established that the standard theory cannot account for the features of simulated Rossby waves propagation, which can only be satisfactorily explained when both the mean flow and bottom topography are accounted for. Flat bottom, no mean flow, modes are completely unable to capture the vertical structure of simulated Rossby wave variability. Irrespective of what the observations tell us, I believe it is pretty clear that the authors' approach cannot tell us anything about actual Rossby waves.

Again – the manuscript does **not** deal with the theory of Rossby waves (be it Trapped or Harmonic)! We only examine the accuracy of the methods used to extract propagation speeds from time-longitude diagrams. Indeed, the manuscript does not "tell us anything about actual Rossby waves" and the reviewer's comment belongs somewhere else and not in a review of the issue our paper addresses.

3. Contrary to what this paper and previous ones assert, theoretical studies of the standard theory based on the WKB approximation are able to account for both the trapping of the Rossby waves as well as for Earth curvature, and it is misleading to refer to such theories as harmonic theories. In WKB theory, one will typically express the pressure anomaly in the form

$$p = A(x,y,z,t)e^{i\Sigma(x,y,t)}$$
$$k = \nabla\Sigma, \quad \omega = -\frac{\partial\Sigma}{\partial t}$$

In such an approach, the amplitude is slowly varying, and will in general decay with latitude, thus capturing the trapped wave behaviour emphasised by the authors. The function $\Sigma$ is a rapidly varying phase function, allowing to define a local wave vector and frequency. Note that a single WKB wave mode is able to represent the observed beta-refraction and a latitudinally varying phase speed.
In contrast, the basic wave mode considered by Paldor's group is separable in latitude, and typically chosen of the form

$$p = A(y)e^{i(kx-\omega t)}$$

Arguably, if the term 'harmonic mode' needs to be used, it seems more appropriate to the modes considered by Paldor's group, since it is clearly what they chose for the temporal and zonal dependence of their mode. As a result, such a mode does not capture the beta-refraction pattern described by Shopf et al. (1981) for instance, raising the question of how useful this kind of mode is to describe mid-latitude Rossby waves.

Though this point has nothing to do with the sermon of our paper we agree with the reviewer. The difference between the Trapped wave theory and the Harmonic wave theory is precisely the form of A(y) in the last expression for $p$: In the traditional, Harmonic, theory the variation of $p$ is sinusoidal so these waves (that spread over the entire latitudinal domain) are named Harmonic while in the Trapped wave theory A(y) has the form of Airy function whose maximum is located near the equatorward boundary (southern in the northern hemisphere). In mid-latitudes Schopf's theory employs the usual "f is constant though its derivative does not vanish" while in his equatorial ray theory the frequency is y-dependent so the concept of separation of variables, that underlies the form of $p$ is entirely lost (d/dy should include the latitudinal derivative of the frequency). Again – we emphasize that these (interesting) issues have nothing to do with the sermon of the present work!

---

## Author Comment (AC2) · 25 Jul 2019

**Author's response to:* Referee #2's comments on "Commonly used methods fail to detect known phase speeds of simulated signals of Sea Surface Height Anomalies" *by* Y. De-Leon and N. Paldor**

Characterizing the properties of oceanic Rossby waves is central to understanding the role of the ocean in the climate as much of the response of the ocean to large-scale forcing is mediated by these waves. Indeed, this issue has attracted considerable attention across the ocean sciences, particularly since the advent of accurate altimetry measurements in 1992 when it became possible, in principle, to observe the signature of Rossby waves at the ocean surface, yet many aspects of such waves remain poorly understood. In particular, it has been found that observed phase speeds derived from altimetry data are systematically faster than the speeds suggested by the theory of Rossby waves. A number of explanations for the disagreement between observations and theory have been proposed, including the effects of the mean zonal flow and bottom topography or the fact that many of the westward-propagating features observed in the altimetry data are, in fact, eddies rather than Rossby waves.

The present study tests the ability of several methods to estimate the phase speed of Rossby waves on simulated data, and finds that such methods very often fail to estimate the true phase speed. The authors then conclude that this is the most likely reason for the differences between observed and theoretical phase speeds. The paper is well written, the figures are mostly adequate and clear, and the experiments designed to assess the skill of the various detection appear to have been conducted appropriately. Unfortunately, although the overall aim of the paper is worth pursuing, some aspects of the paper raise doubts and I do not believe that the results from the performed experiments support the authors' conclusion that "none of the methods is reliable for estimating the phase speed of Rossby waves in the real ocean". The authors are right in concluding that none of the methods is able to estimate the true phase speed in the simulated data, but this conclusion cannot be extrapolated to the observed data since, to the extent that I understand the issue, I don't think the simulated data provides an accurate representation of Rossby waves in the real ocean. In conclusion, I think that the manuscript requires substantial revisions and thus I cannot recommend it for publication as it stands. Details on my main concerns and other minor points are provided below.

Following the reviewer's comments, we will include in the revised manuscript oceanic phenomena other than Rossby waves in which the same radon Transform and 2d-FFT methods are employed. Our findings are relevant to all observations (e.g. near shore dynamics, eddy propagation) where propagation speeds are extracted from time-longitude diagram. Our choice of parameter ranges is drawn from the massive usage of the examined methods in the extraction of Rossby wave phase

speed from time-longitude diagrams of satellite observed SSHA signals.

**Main points:**

1. It is unclear to me from Section2.1 how exactly the simulated data are generated. The authors state that "The values of the phase speeds, C, are uniformly distributed in the 0 to -18 cm/s range". Does that mean that each of the 20 or 50 modes is assigned a different phase speed within that range?

Yes, that's exactly what we did. We will clarify it in the revised version of the manuscript.

Long Rossby waves in the ocean are approximately non-dispersive and so their phase speed is the same at all frequencies. Hence, assigning a different speed to every mode, if this is indeed what is done here, seems unjustified. Could you please clarify how exactly phase speed are ascribed to each mode? How do the results change if the same phase speed is used for all modes?

The emphasis is on "Long" while we include all wavenumbers, long and short, so the waves should be considered dispersive. In the case when all modes have the same phase speed, the 2D-FFT methods still fail in many cases (see the existing remark in the second paragraph of the Discussion) while the Radon transform methods will probably detect the phase speed correctly (we will add a note to this effect in the same paragraph).

Also, the range -18 to 0 cm /s contains some rather extreme values, do you get the same results if the speeds are generated from the range (-10, -2) cm/s?

The range of phase speed we employ is an "envelope" of observed values of Rossby waves. As per the reviewer's suggestion we calculated the detection accuracy of the 4 methods in smaller ranges of frequency and phase speed and the conclusions from these results will be added to the revised version of the manuscript.

In low latitudes the phase speed of Rossby waves can easily exceed 15 cm/sec, and in high latitudes it is less than 1 cm/s. See e.g. Fig. 7 of Killworth et al. in the Journal of Physical Oceanography (1997), attached below. We will omit the words "in mid-latitudes" in the 2nd paragraph of section 2.1.

[Figure]

FIG. 7. The unperturbed fastest long planetary wave speed (in cm s⁻¹) (simply $-\beta/f^2$ times the square of the unperturbed internal wave speed shown in Fig. 6). Contour intervals are nonuniform: 30, 20, 15, 10, 8, 6, 4, 2, 1, and 0 cm s⁻¹ westward. (The values 30 and 0 do not occur, but are added for comparison with later diagrams.) The values are masked within 10° of the equator, where equatorial, rather than long planetary wave, theory should hold.

2. On a similar comment, the theory of Rossby waves indicates that Rossby waves have a maximum frequency, which for the ocean is quite restrictive. For example, no baroclinic Rossby waves with periods shorter than 13 weeks are possible poleward of about 15o latitude. Here, the periods are taken from the range 5 to 200 weeks, which again seems to include some rather extreme values. Could you please provide a reference supporting such high frequencies for observed Rossby waves? How do the results change if you restrict the periods of the Rossby waves to, for example, the range 15 to 100 weeks?

Assume that a typical propagation speed is 5 cm/s and examine a wave with 5000 km wavelength. Then:

$$T = \frac{2\pi}{\omega} = \frac{2\pi}{kC} = \frac{\lambda}{C} \Rightarrow T = \frac{5 \cdot 10^6 \ m}{0.05 \ m/s} = 10^8 \ \text{seconds} \approx 1150 \ \text{days} = 165 \ \text{weeks}$$

Considering the Nyquist frequency constraint our choice of longest period of 200 weeks does not seem to be an over-estimate for Rossby waves. Lower values of C and higher values of λ will yield longer periods.

The lower value of 5 weeks does not differ much from 13 weeks. However, as stated our response to comment point #1 above, we will include a brief description of the results for smaller ranges of both frequency and phase speed.

3. Theoretical phase speeds are not only different from observations, they are systematically slower. If the simulated data were an accurate representation of the

real ocean and the detection methods were really the issue here, then the authors should also find a systematic bias in the estimated phase speed. However, there is no mention of this in the paper. The authors only state that all methods fail to estimate the true phase speed of Rossby waves. Do you find any systematic biases? Could you please further elaborate on this?

Right, the observed speeds are always faster than the harmonic speeds but have no systematic bias compared to the Trapped wave's speeds. A clear example of this behavior is given in the comparison shown in Fig. 5 of De-Leon and Paldor, 2017b (reproduced below). The red curve is the Trapped wave speed and the Green curve – the Harmonic speed. Symbols are the observational speeds that are distributed systematically **above** the harmonic speed but with no obvious bias compared to the trapped wave speed.

In addition, we don't state "…that all methods fail to estimate the true phase speed of Rossby waves", but that they fail to estimate a dominant input phase speed regardless of its physical origin i.e. Rossby waves are an example.

[Figure]

Figure 5. The observed phase speeds and the two theoretical phase speeds (trapped and harmonic) as a function of $\phi_m$ in intervals of 0.5° latitude. Blue dots denote latitudes where the estimates of at least two methods agreed by 10 % or less, triangles denote latitudes where such estimates agreed by 11 to 12 % and squares denote latitudes where the agreement is 25 %. No reliable estimates were obtained north of 35° S and in some more latitudes. The sum of squares of the distances in (cm s$^{-1}$)$^2$ between trapped wave phase speeds and observed speeds (3.5) is much smaller than that of harmonic phase speeds (15.3).

4. In assessing the skill of the various methods, the authors assign a score of ½ if the dominant mode falls in one of its nearest neighbors. This seems to me like a rather arbitrary choice. Why not the second nearest neighbor or the third one? Can you estimate a "standard error" for the phase speed estimates based on the multiple realizations and assign a score of 1 when the true value is within one standard error and zero otherwise? This would be, in my view, a fairer metric for skill. Also, I think that 50 realizations is not sufficient and would suggest you use at least 100, if not 1000.

Indeed, our choice is arbitrary but so is any other choice. We will emphasize it in the revised text. The number of cases where the detected mode was 1-bin away from the dominant input mode (i.e. the score was ½) is very small in all signals we examined. As for the number of realizations, we didn't find significant difference between 25, 50 or 100 repeats.

**Minor points:**
Page 1. The spatiotemporal resolutions quoted here for the altimetry data refer to the grid size and time step of the altimetry gridded products rather than the scales that can actually be resolved by altimeters. Depending on latitude, the spatial separation between altimetry tracks can be of several hundred kilometers and altimeters take measurements over the same location once every 10 days at most. I think that some clarification is needed here, along with some references.

We define the grid in the same way it is defined by Aviso in their description of the altimetry gridded products they distribute to the community.

Page 1. "these features propagate..." What features? Please clarify. We removed this sentence.
Page 1. "Rossby waves that propagate westward" I suggest you remove "that propagate westward" as this seems redundant in this particular sentence. We removed this sentence.
Page 1. replace "diagrams at certain latitude" with "diagrams at a certain latitude". Done.
Page 2. "phase speed exceeds". We removed this sentence.
Page 2. I suggest "in the -18 to 0 cm/s range". Done.
Page 5. I suggest "None of the methods can identify a dominant input ..." Done.

---

## Author Response (AR2)

**Authors' response to reviewer #1 comments on OS-2019-34-R1**

Authors' response to the individual comments is denoted in red

Report 1
First, I want to thank the authors for their response to my previous comments. Overall, the authors have suitably addressed my comments and clarified the points I raised. While I still think that this paper is largely an academic exercise, it is scientifically valid (assuming the analysis has been properly done), and so it is not my place to hold this paper up. I have only one remaining important comment and a few minor ones.

The authors dismiss my comment about the arbitrariness of the metric used to assess the accuracy of the various detection methods by saying that "Indeed, our choice is arbitrary but so is any other choice". The authors' response is rather unscientific and I disagree with it. Choices about skill metrics should be governed by logical rather than arbitrary considerations and they should be clearly justified, otherwise the results will be suspect. On a related comment, I think that rather than saying whether or not a method detects the true propagation speed based on an arbitrary metric, it would be far more useful to provide a measure of the uncertainty associated with estimates of propagation speed.

We consider another scoring "yardstick": 1 at the bin, 2/3 at the nearest neighbors, 1/3 at the next to nearest neighbors and 0 away. The statistics of success of the 4 methods differs only slightly (up to 3%) from the "yardstick" we originally employed (which was 1/2, 1, 1/2). Despite the statistical (perhaps even somewhat arbitrary) of our examination it underscores the unreliable nature of the currently employed methods in detecting the phase speed of actual/observed signals.

The authors write "Indeed, our choice is arbitrary but so is any other choice. We will emphasize it in the revised text.". However, I cannot see where this has been emphasized in the text.

Right, the issue is addressed in the revised version (see L. 14-16 on P. 4 and L. 10-11 on P. 5)

Similarly, the authors write "As for the number of realizations, we didn't find significant difference between 25, 50 or 100 repeats." This is not mentioned in the paper and I think it should.

The issue is addressed in the revised version where 25 and 100 were added to the original repeats of 50 (see L. 9 on P. 5)

**Authors' response to reviewer #2 comments on OS-2019-34-R1**

Authors' response to the individual comments is denoted in red

Report 2
I rather struggle to see the point of this paper; it seems to be setting up an implausible strawman problem and then shooting it down.

The sermon of this paper is that no currently employed method (and we examine more than just the Radon method) can be considered reliable in detecting the correct phase speeds of observed signals in the ocean. To establish the inaccuracy of these methods we use artificial signals with known phase speeds that mimic in some general way (more than 1 amplitude and many phase speeds) oceanic signals. There is no "strawman" in our problem – just an artificial signal that examines whether these methods can detect the known phase speed in an artificially generated "signal". If the reviewer knows of a more fitting test s/he should have pointed it to the community since as it now stands the reviewer's approach does not permit any examination of a proposed method (in any field!) since, by definition, such an examination implies the application of a known speed to an artificial signal in order test the detection accuracy of the method being examined.

The idea behind the radon transform is that, in situations where there is a dominant propagation speed, it identifies that speed as a direction in t-x space along which the variance of the mean signal is maximised. The emphasis is on "dominant speed". This requires that there be either a single mode propagating at that speed with squared amplitude larger than the sum of the squares of the amplitudes of all other modes, or that there be a cluster of modes with similar speeds sufficient to be dominant over the modes with different speeds. The former case clearly requires the single mode to have a much larger amplitude than all the other modes. That seems to be what this paper is showing (also from Figure 5, we see how the "random" choice of speeds for other modes will tend to produce a clustering of energy at slower speeds: the result will depend on how the randomly chosen modes happen to cluster as a function of theta, with different choices giving different results).

The reviewer might be right in suggesting that when "a single mode propagating at that speed with squared amplitude larger than the sum of the squares of the amplitudes of all other modes" there is a dominant mode (as shown in our Fig. 3b for N=20 and amplitude of 5 for the dominant input mode so $5^2 = 25 > 19 \times 1$). However, in the real ocean it is impossible to determine the relevant amplitudes and the number of modes (N).

The "cluster of modes with similar speeds" the reviewer seeks is represented in our study by the width of the bin i.e. all modes in each bin are assumed to have the same amplitude. The addition of internal amplitude structure in each bin will not change the statistics we compute.

It is hard to think of a physical system for which this extreme case is a plausible model. If there is propagation, there is likely to be a smooth variation of speeds as a function of frequency and wavenumber. The aim with radon transforms as applied to Rossby waves is usually to identify the long Rossby wave speed, which is independent of frequency and wavenumber and hence will apply to a range of modes (and hence a clustering of energy at a particular angle in t-x space), with the shorter waves becoming more dispersive and thus producing a smaller signal in the

radon transform. It is arguable whether this is really achieved (see a wide range of papers recently with Dudley Chelton, among others, arguing that the dominant speed tends to be more related to nonlinear, coherent eddies). Even in the purely linear limit, the radon transform clearly isn't the appropriate tool if interest lies in the dispersive part of the wave spectrum. In that case a two-dimensional Fourier method seems a good place to start, together with the idea of searching for a dispersion relation rather than simply a single speed. For example, see Zang and Wunsch (JPO 1999, doi: 10.1175/1520-0485(1999)029<2183:TODRFN>2.0.CO;2).

We are unsure of the relevance of this point to our study. The dispersive nature of the waves has nothing to do with the method of detection! Any examination of a widely used method requires the generation of a simulated signal in which the dominant phase speed is known (otherwise the method can't be tested) and the only degree-of-freedom is the degree of dominance of this mode. This is precisely what we do in our study. Our examination does not address waves versus eddies and/or linear versus nonlinear features and/or dispersive versus non-dispersive waves. It's just an assessment of the methods under conditions where the "answer" is known from the outset. Likewise, we don't understand the reviewer's reference to the Zang and Wunsch (1999) paper since we also examine the 2D-FFT method.

So there seems to me to be little added value in this contribution. The radon transform remains a good way to identify a dominant speed when there really is one. When there isn't, it is a poor method. There is nothing new or controversial about that.

Obviously, if one accepts the performance of existing methods as satisfactory there's nothing that needs to be examine. We, however, feel that critical assessment of performance are at the heart of scientific quest.

[revised manuscript text omitted]

---

## Author Response (AR3)

[revised manuscript text omitted]

Authors' response to editor's comments on os-2019-34-R2

In the revised version both points are addressed in the Discussion section on P. 6.

The first point (the difficulty with phase speed distribution in the ocean) is addressed in lines 15-21.

The second points (application to dominant input mode that's made up of a cluster of modes with close phase speeds) is addressed in lines 22-33.
Briefly, the additional calculation we performed with a symmetric "cluster" of 3 randomly selected, adjacent, modes (that have similar phase speeds) with amplitudes (2,4,2) has demonstrated that in all methods clustering slightly increases the success in detecting the dominant input mode. It is unclear whether this improvement in detection results from the wider range of input modes (3 bins compared to 1 while keeping the maximal amplitude nearly identical) only in a finite range of phase speed and whether the same effect will also hold in the ocean where the range of phase speeds is unbounded.